# OpenReview forum: "Generative Modeling of Molecular Dynamics Trajectories"
_NeurIPS.cc/2024/Conference — NeurIPS 2024 poster_

### Official Review · Reviewer_b8Jp · 2024-07-10

**Soundness:** 3
**Presentation:** 2
**Contribution:** 2
**Rating:** 7
**Confidence:** 2

**Summary:**

The paper suggests a flow-based generative framework on molecular trajectories, with various downstream tasks such as forward simulation and transition path sampling. Additionally, the model is trained in a transferable setting, across tetrapeptides.

**Strengths:**

1. Extensive experiments over various downstream tasks
2. Transferable settings for tetrapeptides

**Weaknesses:**

1. Experiment baselines

The baselines of experiments are mostly the Markov State Models. I think it would also be good if there were some comparison between other models, though I understand that many prior works targeted Alanine dipeptide not tetrapeptides.
- Forward simulation: ITO$^{[1]}$, Timewarp$^{[2]}$

- Interpolation (Transition path sampling): PIPS$^{[3]}$

2. (Minor) Necessity of additional tasks

The necessity of additional tasks relatively seems weak compared to tasks such as forward simulation, TPS, specially the inpainting design. Rather than additional tasks, ablations for stability might be a better? One can obviously see that scaling to long trajectories shows the stability against the time scale, and protein simulation shows the stability against space complexity.

**Minor typos, suggestions**

- Definition of S-MPNN only exists in the Appendix. It would great to point out that more details are in the appendix, in the first paragraph of section 4.4
- Figure 6 is not referenced in the main paper, only the appendix
- Figure 2F, reference of that blue indicates the side chains and orange indicates the backbones seems to be missing

[1] Implicit transfer operator learning: Multiple time-resolution surrogates for molecular dynamics, NIPS 2023

[2] Timewarp: transferable acceleration of molecular dynamics by learning time-coarsened dynamics, NIPS 2023

[3] Stochastic Optimal Control for Collective Variable Free Sampling of Molecular Transition Paths, NIPS 2023

**Questions:**

1. Difference between downstream tasks

Could the upsampling task be seen as a superset of interpolation? Since upsampling with two given frames would the same as interpolation.

2. Training on one tetrapeptide

Just curious, though the authors has presented a transferable setting, are there any results when the model is trained for a specific tetrapeptide and tested on downstream tasks?

**Limitations:**

1. Unconditional generation

As the authors have mentioned, unconditional generation is impossible since the model relies on key frames.

---

> ### Author Rebuttal · Authors · 2024-08-03
>
> Thank you for the review! To address your questions and concerns:
>
> ---
>
> **Experimental baselines**
>
> We now provide new results comparing our work to Timewarp and ITO. Emphatically, these comparisons are limited to the forward simulation task as Timewarp and ITO are not capable of solving the other tasks.
>
> * For **Timewarp**, we use the 4AA model with weights from the authors and sample 100 ns trajectories by running 2000 inference steps with timestep 50 ps. We do not use MH acceptance steps as the authors found exploration of the energy landscape to be much more effective without them.
> * For **ITO**, transferable models across tetrapeptides are not available. We, therefore, train ITO ourselves on our tetrapeptide dataset with timesteps of 500 ps. We then run 200 inference steps to sample 100 ns trajectories.
>
> For both methods, we observe that trajectories are unstable without further intervention (see **Figure 2** of the PDF attached to the global response). To bolster the baselines, we run OpenMM relaxation steps between each timestep to proceed with further analysis. We note that the Timewarp authors, in lieu of relaxation, rejected steps with an energy increase of 300 kJ / mol; however, we found that this strategy would reject the majority of proposed steps on generic tetrapeptides.
>
> In **Figure 3** of the PDF attached to the global response, we visualize the free energy surfaces and torsion angle distributions for several peptides from Timewarp and ITO compared with MDGen. Our model obtains much better results across the board.
>
> Quantitatively, we obtain the following Jensen-Shannon divergences from the forward simulation rollouts (c.f. Table 2):
>
> |C.V.| MDGen | Timewarp | ITO |
> |:-| :-: |  :-: | :-: |
> |Torsions (bb)| **0.130** | 0.325 | 0.564 |
> Torsions (sc) | **0.093** | 0.427 | 0.462 |
> Torsions (all) | **0.109** | 0.383 | 0.505 |
> TICA-0 | **0.230** | 0.265 | 0.538 |
> TICA-0,1 joint | **0.316** | 0.419 | 0.756 |
> MSM states | 0.235 | **0.222** | 0.414 |
> Runtime (s)| **60** | 599 | 2083 |
>
> To evaluate the dynamics, we also compare the Pearson correlation of predicted torsional relaxation times (c.f. Figure 2F). We could not compute these quantities for ITO as the vast majority of torsions did not decorrelate within the ITO rollouts.
>
> |C.V.| MDGen | Timewarp | ITO |
> |:-| :-: |  :-: | :-: |
> Torsion relaxation (bb)| **0.55** | 0.04 | ---|
> Torsion relaxation (sc) | **0.97** |  0.77 | ---|
>
>
> Altogether, the results confirm that MDGen is more successful and efficient at modeling thermodynamics and kinetics than Timewarp or ITO.
>
> **PIPS** We note that PIPS is a method for steering simulations towards a desired target state, whereas MDGen aims to learn from existing unsteered simulations in a transferable manner to sample transitions of unseen systems. Due to the difference in tasks, a direct comparison with PIPS is difficult to formulate.
>
> **Additional tasks vs ablations for stability**
>
> A central point of our paper is to highlight the novel capabilities afforded by our trajectory modeling framework. We chose several tasks out of reach of previous methods, and hence less studied in ML papers, but all of which we believe are scientifically meaningful. With that said, we acknowledge that different tasks will appeal to different readers, and appreciate your input about their relative importance.
>
> We are happy to consider suggestions for further experiments for stability (time permitting) beyond the existing experiments you referenced.
>
>
> **Difference between the downstream tasks**
>
> Yes, upsampling could be seen as a superset of interpolation in terms of the conditioning input. However, their conceptual aims are somewhat different: In the interpolation setting, we choose endpoints in different (often distant) macrostates of the energy landscape to study how they are connected. On the other hand, in upsampling the conditioning frames are chosen based on timestep, irrespective of whether they exhibit interesting transitions.
>
>
> **Training on one tetrapeptide**
>
> Apart from the Hyena experiment (Section 4.4), we opted to focus on the transferable setting as we believe this is the arena in which ML-based MD emulators will be useful in practice. Additionally, since our model generates 10 ns at a time, we do not anticipate that learning from a single 100 ns trajectory would be meaningful, as this amounts to only 10 statistically independent training examples.
>
> **Unconditional generation**
>
> While it is true that the use of key frames impacts the ability to do unconditional generation, we nevertheless opted for this design choice as unconditional generation of trajectories is (to our knowledge) not a problem of scientific interest. On the other hand, the key frames allow the conditional modeling problems to be consideribly simplified and clarified, leading to the strong results shown in our experiments. Hence, the use of key frames could instead be considered a technical insight of our model that enables it to contribute to the development of real-world, impactful scientific problems, at the cost of problems of lesser interest.
>
> **Minor typos, suggestions**
>
> Thank you for noting these; we will incorporate them in the revision!
>
> ---
>
> We hope the new discussion and results address your concerns! Please let us know if there are further opportunities to improve the score.

---

> > ### Comment · Reviewer_b8Jp · 2024-08-08
> >
> > I thank the authors for the detailed response and additional experiments. Additional experiments on forward simulation shows better performance of MDGen over Timewarp and ITO, where prior works fail in longer simulation time. All questions have been resolved, and I have raised my score accordingly.

---

### Official Review · Reviewer_hVZ7 · 2024-07-10

**Soundness:** 3
**Presentation:** 3
**Contribution:** 3
**Rating:** 7
**Confidence:** 4

**Summary:**

The authors propose MDGen -- a generative model to sample molecular dynamics trajectory conditioned on key frames. This is a direct application of video generation techniques to solve domain challenges in protein modeling. Specifically, SiT and flow matching models are used to sample SE(3)-invariant representation of all-atom protein representations. This work demonstrates the effectiveness of MDGen primarily on tetrapeptide systems, where the authors showcase four downstream application tasks including forward simulation, interpolation, upsampling, and dynamics-conditioned inpainting.

In general, I find this manuscript well-written and easy to follow. The model performance looks reasonable on tetrapeptides, yet the results are proof-of-concept in nature and generalization to larger proteins remain challenging. However, it is one of the pioneering work in AI protein modeling to directly emulate MD simulation trajectories using data-driven approaches. To that end, I think it would be beneficial for this work to gain visibility across the research community to inspire future studies.

**Strengths:**

- It is one of the pioneering work to adopt video generation techniques for MD trajectory generation. Although conceptually straightforward, good practices to generate time-coherent MD trajectories across different protein systems remain underexplored.
- The authors demonstrated a variety of downstream tasks using the same model architecture. The underlying modeling framework seems versatile and transferrable across different applications..
- Performance benchmark and analysis on tetrapeptides are comprehensive and provides insights to modeling these peptide systems.
- I think it is a good idea to model residue offsets relative to the key frames in order to bypass the need to learn sequence-to-structure mapping. MD simulations always start from a seed structure, so I do not think this is a key limitation as mentioned in L#310-312.

**Weaknesses:**

- Benchmark and evaluation results on tetrapeptides, although comprehensive, are proof-of-concept in nature. It may not be sufficient to demonstrate transferability to general protein systems.
- Performance on ATLAS (i.e., larger proteins instead of short peptides) does not seem promising. MDGen performance is worse than AlphaFlow in Table 4. I wonder if the main bottleneck is training data quality/availability, or model architecture?

**Questions:**

- L#141, when $K > 1$, how to ensure roto-translation prediction consistency across $K$ key frames and obtain a final $\hat{g}_j$?
- Table 2, with 100 ns being the ground truth, the non-zero JSD in the last column originates from subsampling the simulation trajectory?
- Figure 2F. My understanding is that sidechains exhibit faster dynamics while backbone motions are slower. The low correlation for backbone suggests that MDGen is not good at learning slower dynamics, which are typically more interesting to researchers?
- Temporal coherence between generated protein conformations is mainly evaluated using auto-correlation in this work. Is it possible to show other metrics to capture detailed structural quality and variation during time evolution?
- Why is MDGen more effective at sequence recovery than MPNN? More explanation and analysis would be helpful here.
- Would it be possible to emulate MD simulation trajectory of the 12 fast folding proteins from [Shaw 2009](https://dl.acm.org/doi/abs/10.1145/1654059.1654126)? They are smaller than ATLAS proteins and longer than tetrapeptides, with much longer simulation time and rich dynamics.
- It would be nice to see if MDGen could infer a trajectory given an [apo/holo pair](https://arxiv.org/abs/2304.02198).

**Limitations:**

See weaknesses and questions.

---

> ### Author Rebuttal · Authors · 2024-08-03
>
> Thank you for the review! To address your questions and concerns:
>
> ---
> **Limited to tetrapeptides**
>
> We have focused on tetrapeptides as model systems in this work for two key reasons:
> * We can run simulations for thousands of systems in order to properly test the generalization abilities of our model.
> * We can build Markov State Models for each system, allowing the careful benchmarking for forward simulation and interpolation capabilities.
>
> Both of these aspects are important for the thorough, careful benchmarking of the new capabilities we demonstrate. We opted to prioritize these careful studies to support the core claims of our work, in lieu of expanding its scope to larger and more diverse systems, which we think are best left to future work.
>
> **Comparison with AlphaFlow**
>
> It is true that the results on ATLAS simulations are worse than AlphaFlow. However,
> * MDGen learns the harder task of providing temporal consistency across structures, which is beyond the capabilities of AlphaFlow. However, our metrics are obtained from the AlphaFlow paper and only assess the ensemble similarity of unordered sets of conformations, favoring their method.
>     * To more concretely demonstrate learned temporal consistency, we now show an MDGen trajectory connecting apo and holo states of adenylate kinase in **Figure 4** in the PDF attached to the global response.
> * AlphaFlow is a much larger model with significant transfer learning from O(10^5) structures via AlphaFold, and has a much better pretrained understanding about likely protein conformations. On the other hand, MDGen is trained from scratch on only O(1000) proteins.
>
>
>
> With that said, we do not anticipate that this will be the final, definitive architecture for trajectory modeling of protein systems. Rather, our experiments serves as a proof-of-concept that these capabilities _can_ be demonstrated on proteins, even with the extremely limited available data. We anticipate that a combination of expanded datasets and further architectural exploration can improve results on ATLAS in future work.
>
> **Additional structural quality and variation metrics**
>
> We note that in addition to autocorrelation, our torsion angle distributional metrics and free energy surfaces also assess the quality of structures in the trajectory. With that said, we are happy to report addition metrics as requested.
>
> To more stringently assess structural quality in MDGen forward simulation rollouts, we compute the distributions of
> * The closest distance between nonbonded atoms
> * Nonbonded energy (Coulomb + Lennard-Jones)
> * Torsional energy
> * Heavy atom bond lengths
> * Radius of gyration
>
> These distributions are shown and compared to the ground truth in **Figure 1** in the PDF attached to the global response. We find that the vast majority of MDGen structures are of high quality (i.e., clashes are rare) and adhere closely to the ground truth distributions.
>
> To assess structural variation across time, we report the aligned RMSD between frames spaced at regular intervals, and compare to the same metric computed from the reference simulation. This gives an idea of how fast the structures should be moving. We observe that the trajectories have dynamics that closely resemble the ground truth.
>
> | Interval | MDGen | Reference MD |
> |:-| :-: | :-: |
> | 10 ps | 1.02 A | 0.99 A |
> | 20 ps | 1.19 A | 1.17 A |
> | 50 ps | 1.45 A | 1.44 A |
> | 100 ps | 1.65 A | 1.67 A |
> | 200 ps | 1.86 A | 1.89 A |
> | 500 ps | 2.11 A | 2.18 A |
> | 1000 ps | 2.28 A | 2.37 A |
>
> Please let us know if you had other metrics in mind; we are happy to analyze the structures further!
>
> **Other questions**
> >L#141, when K>1, how to ensure roto-translation prediction consistency across K key frames and obtain a final g^j?
>
> We predict all-atom coordinates using each set of key frames (and corresponding predictions) and average these coordinates.
>
> >Table 2, with 100 ns being the ground truth, the non-zero JSD in the last column originates from subsampling the simulation trajectory?
>
> Similar to the other columns, the MD baselines indicate replicate simulations, i.e., an independent 100 ns simulation.
>
> >Figure 2F. My understanding is that sidechains exhibit faster dynamics while backbone motions are slower.
>
> It is true that the slower dynamics are typically more interesting; unsurprisingly, they are also harder to simulate and learn. Our method does not perfectly recover these dynamics, but it is the first of its kind to report good results on this task.
>
> >Why is MDGen more effective at sequence recovery than MPNN? More explanation and analysis would be helpful here.
>
> MDGen is trained to use more input information about the peptide, namely the intermediate dynamics of the unmasked residues. Our inverse folding baselines are not able to make use of these partially observed intermediate structures.
>
> >Would it be possible to emulate MD simulation trajectory of the 12 fast folding proteins from Shaw 2009?
>
> The DESRES trajectories are unusual in that they were simulated at elevated temperatures to invoke unfolding events. In terms of absolute displacement, these are vastly larger motions away from the starting structure than those seen in ATLAS simulations. As such, we do not think our architecture, with its dependence on key frames, would be optimal for modeling such trajectories. Earlier in the project, we considered implementing a distogram-based architecture for the DESRES proteins, but opted to focus on the transferable setting with ATLAS instead.
>
> >It would be nice to see if MDGen could infer a trajectory given an apo/holo pair.
>
> Thanks for the suggestion! We have newly trained an interpolation model on ATLAS data and visualize a trajectory between the apo and holo states of adenylate kinase (1AKE / 4AKE) in **Figure 4** of the PDF attached to the global response.
>
> ---
>
> We hope the new discussion and results address your concerns! Please let us know if there are further opportunities to improve the score.

---

> > ### Comment · Reviewer_hVZ7 · 2024-08-12
> > **Response to Rebuttal**
> >
> > Dear authors,
> >
> > Thanks for your response. Most of my concerns have been properly addressed. I think this pioneering work on generative modeling for protein/peptide dynamics should be shared with the community to inspire follow-up work. I have raised my scores accordingly.

---

### Official Review · Reviewer_zaG3 · 2024-07-12

**Soundness:** 3
**Presentation:** 3
**Contribution:** 2
**Rating:** 4
**Confidence:** 4

**Summary:**

The paper presents a new framework for generating trajectory of molecular geometries, ie, generative modeling for molecular dynamics. The paper proposes tokenization methods to tokenize the trajectory and learn flow models on the data. Experiments demonstrate the effectiveness of several tasks including forward sampling,  interpolation, and up sampling.

**Strengths:**

1. The paper tackles a new problem in molecular dynamics generation, which has not been explored in existing literature.

2. The paper is in good structure and easy to follow.

3. The paper provides a detailed analysis of several domain tasks on interested molecular structures, which demonstrate the critical usage in some scenarios.

**Weaknesses:**

1. Limited ML technical contribution, as all components exist in previous molecular generative models.

2. The experiment is comprehensive from a domain perspective. However, I feel the experiments lack some benchmarking comparison with state-of-the-art molecular generative models for related tasks. See my question below.

**Questions:**

I think existing methods can also tackle several tasks. For example, for the forward sampling task, previous generative MD models like Timewarp (Klein et al., 2024) and ITO (Schreiner et al., 2024) can also be used for the task. A numerical comparison with these baselines can help to justify the effectiveness of the proposed method.

**Limitations:**

Limitation is nicely discussed in the paper.

---

> ### Author Rebuttal · Authors · 2024-08-03
>
> Thank you for the review! To address your questions and concerns:
>
> ---
>
> **Limited ML technical contribution**
>
> In writing the paper, we opted to placed more emphasis on the experimental results. Nonetheless, we respectfully disagree that our work has limited ML technical contribution.
>
> We highlight that the following points are novel from a modeling perspective and, to our knowledge, have not been used in any prior molecular generative model:
> * The reduced representation of residue all-atom coordinates into $\mathbb{R}^{21}$.
> * The use of key frames to obtain SE(3)-invariant tokens from molecular trajectories.
> * The use of scalable vanilla transformers for molecular generation while respecting the problem symmetries (i.e., without breaking equivariance like molecular conformer fields or AF3).
>
> Indeed, as we write in the Introduction, our approach provides a novel strategy for how to circumvent the expensive architectures normally used for equivariant molecular modeling, a technical contribution that could be useful for many future works.
>
> Additionally, we believe that the insight of using trajectory generation to solve inverse problems like interpolation or upsampling in itself constitutes methodological novelty, akin to the novelty appreciated in the many surprising applications of diffusion models in RL.
>
>
>
> **Comparison with Timewarp and ITO**
>
> We now provide new results comparing our work to Timewarp and ITO. Emphatically, these comparisons are limited to the forward simulation task as Timewarp and ITO are not capable of solving the other tasks.
>
> * For **Timewarp**, we use the 4AA model with weights from the authors and sample 100 ns trajectories by running 2000 inference steps with timestep 50 ps. We do not use MH acceptance steps as the authors found exploration of the energy landscape to be much more effective without them.
> * For **ITO**, transferable models across tetrapeptides are not available. We, therefore, train ITO ourselves on our tetrapeptide dataset with timesteps of 500 ps. We then run 200 inference steps to sample 100 ns trajectories.
>
> For both methods, we observe that trajectories are unstable without further intervention (see **Figure 2** of the PDF attached to the global response). To bolster the baselines, we run OpenMM relaxation steps between each timestep to proceed with further analysis. We note that the Timewarp authors, in lieu of relaxation, rejected steps with an energy increase of 300 kJ / mol; however, we found that this strategy would reject the majority of proposed steps on generic tetrapeptides.
>
> In **Figure 3** of the PDF attached to the global response, we visualize the free energy surfaces and torsion angle distributions for several peptides from Timewarp and ITO compared with MDGen. Our model obtains much better results across the board.
>
> Quantitatively, we obtain the following Jensen-Shannon divergences from the forward simulation rollouts (c.f. Table 2):
>
> |C.V.| MDGen | Timewarp | ITO |
> |:-| :-: |  :-: | :-: |
> |Torsions (bb)| **0.130** | 0.325 | 0.564 |
> Torsions (sc) | **0.093** | 0.427 | 0.462 |
> Torsions (all) | **0.109** | 0.383 | 0.505 |
> TICA-0 | **0.230** | 0.265 | 0.538 |
> TICA-0,1 joint | **0.316** | 0.419 | 0.756 |
> MSM states | 0.235 | **0.222** | 0.414 |
> Runtime (s)| **60** | 599 | 2083 |
>
> To evaluate the dynamics, we also compare the Pearson correlation of predicted torsional relaxation times (c.f. Figure 2F). We could not compute these quantities for ITO as the vast majority of torsions did not decorrelate within the ITO rollouts.
>
> |C.V.| MDGen | Timewarp | ITO |
> |:-| :-: |  :-: | :-: |
> Torsion relaxation (bb)| **0.55** | 0.04 | ---|
> Torsion relaxation (sc) | **0.97** |  0.77 | ---|
>
>
> Altogether, the results confirm that MDGen is more successful and efficient at modeling thermodynamics and kinetics than Timewarp or ITO.
>
> ---
>
> We hope the new discussion and results address your concerns! Please let us know if there are further opportunities to improve the score.

---

### Official Review · Reviewer_b3SL · 2024-07-14

**Soundness:** 3
**Presentation:** 3
**Contribution:** 3
**Rating:** 6
**Confidence:** 3

**Summary:**

In this work, the authors proposed MDGen, a new framework that aims to model molecular dynamics trajectories via generative modeling techniques. By properly encoding the Protein MD trajectories according to the characteristics of key frames, MDGen adopts flow matching techniques (both continuous and discrete flow matching) to generatively model MD trajectories. As a unified framework, MDGen is able to perform diverse tasks including forward simulation, interpolation, upsampling and inpaiting. Extensive experiments are conducted to demonstrate the effectiveness of MDGen.

**Strengths:**

1. The problem this work aims to tackle is of great significance in scientific domains lie computational biology.
2. The formulation of molecular (protein) trajectories by using key frame references is reasonable and compact for reducing the modeling difficulties.
3. The experiments are comprehensive.
4. The paper is well-written and easy to follow.

**Weaknesses:**

1. Lack of discussion on related works. This work does not discuss related works on the same topic. Some works are mentioned in the Introduction section, but I still recommend that there should be an independent Related Works section for comprehensive discussion.  Here are also several works that are worth discussing: (1) EGNO, which uses neural operator learning approach to also model the trajectory dynamics of molecules; (2) DiffMD, which uses diffusion models to simulate molecular dynamics. The quality of this work should be further improved if the authors could carefully discuss the differences between MDGen and these works and the strengths of MDGen compared to these works.

2. Lack of ablation studies. MDGen is composed of several parts, including the design of the backbone model, the design choices of flow matching framework, and the adoption of Hyena architecture for efficiency consideration. In addition to the aimed tasks, it would further improve the quality of this work if the authors could conduct ablation studies on these aspects to help readers know what the influence of each part of MDGen is.

**Questions:**

N/A

**Limitations:**

The authors carefully discuss the limitations of this work.

---

> ### Author Rebuttal · Authors · 2024-08-03
>
> Thank you for the review! To address your questions and concerns:
>
> ---
>
> **Further discussion on related works**
>
> Thanks for mentioning these works. We are happy to discuss them in the revision alongside the existing related work in the Background which we will also expand as per your suggestion. Briefly:
>
>
> * EGNO is a fourier neural operator that, given a molecular structure and a time delta, *deterministically* predicts the structure after the time delta. The time delta is constrained to be between 0 and a maximum time which in practice is 5fs - this is 20,000,000 smaller than the duration of our trajectories in forward simulation. Notably, the possible structures after a time delta (in the absence of the initial velocity) are a distribution, which EGNO fails to model since it is not a generative model.
> * DiffMD adds noise to molecular structures and denoises them to generate a future structure. Repeating the process autoregressively yields a trajectory.
>
> When producing trajectories of timescales similar to MDGen, both methods produce their trajectories autoregressively which limits them to forward simulation as an application and prevents past frames from being informed by future frames. Meanwhile, MDGen generates all frames in a trajectory jointly, which can improve temporal coherence and enables addressing additional tasks such as interpolation, upsampling, and inpainting.
>
>
> **Lack of ablation studies**
>
> We now provide new experimental results ablating components of the method. The ablations are as follows:
>
> * **No IPA embedding**: the model is not provided IPA embeddings of the key frames and amounts to a vanilla Scalable Interpolant Transformer
> * **No $SE(3)$ invariance**: there is no SE(3)-invariant tokenization and the model operates directly over the frame representations $g_j$ rather than frame offsets $g_i^{-1}g_j$
> * **No frames/torsions**: the model operates directly over all-atom coordinates rather than the reduced representation in $\mathbb{R}^{21}$
>
> We run these ablations on the forward simulation experiment and obtain the following Jensen-Shannon divergences (c.f. Table 2). For a fair comparison, we use the baseline model after training for the same number of epochs as the ablations. The ablations all perform worse than the baseline model.
>
> |C.V.| Baseline | No IPA embedding | No $SE(3)$ invariance | No frames/torsions |
> |:-| :-: |  :-: | :-: | :-: |
> |Torsions (bb)| **0.161** | 0.339 | 0.195 | 0.537 |
> Torsions (sc) | **0.106** | 0.249 | 0.262 | 0.502 |
> Torsions (all) | **0.130** | 0.287 | 0.233 | 0.517 |
> TICA-0 | **0.245** | 0.375 | 0.298 | 0.510 |
> TICA-0,1 joint | **0.345** | 0.500 | 0.416 | 0.657 |
> MSM states | **0.237** | 0.250 | 0.278 | 0.408 |
>
> ---
>
> We hope the new discussion and results address your concerns! Please let us know if there are further opportunities to improve the score.

---

> ### Comment · Reviewer_b3SL · 2024-08-10
>
> Thank you for your clarifications. Most of my concerns have been addressed. I choose to keep my positive rating.

---

### Official Review · Reviewer_TTLp · 2024-07-20

**Soundness:** 3
**Presentation:** 3
**Contribution:** 2
**Rating:** 5
**Confidence:** 3

**Summary:**

The paper presents a novel generative model for molecular dynamics (MD) trajectories called MDGEN. This model aims to serve as a flexible surrogate for MD simulations by generating entire trajectories conditioned on initial frames. It addresses tasks such as forward simulation, transition path sampling, trajectory upsampling, and dynamics-conditioned molecular design. The model is evaluated on tetrapeptide simulations and demonstrates its capability to generate reasonable ensembles of protein monomers.

**Strengths:**

Novelty and Scope: The approach introduces a novel paradigm for surrogate modeling of MD, extending the capabilities of existing models to handle a variety of tasks that are not straightforward with current methods.
Generative Framework: The use of generative modeling for entire MD trajectories is a significant advancement, as it allows for a broader range of applications including forward and inverse problems.
Comprehensive Evaluation: The paper evaluates MDGEN on several tasks, demonstrating its effectiveness in forward simulation, interpolation, upsampling, and inpainting. The results show promising performance in terms of distributional similarity, dynamical content, and computational efficiency.
Technical Implementation: The detailed description of the tokenization process and the flow model architecture provides a clear understanding of how the model operates. The use of SE(3)-invariant tokens and the scalable interpolant transformer (SiT) backbone are well-motivated choices.

**Weaknesses:**

Complexity and Accessibility: The model’s complexity might pose challenges for reproducibility and accessibility for researchers who are not deeply familiar with both molecular dynamics and advanced generative modeling techniques.
Evaluation on Larger Systems: While the paper provides proof-of-concept evaluations on proteins, the primary focus remains on smaller tetrapeptides. The model's scalability and effectiveness on larger and more complex molecular systems need further exploration.
Dependence on Key Frames: The reliance on key frames for conditional generation limits the model’s ability to perform unconditional generation or inpainting of residue roto-translations, which could be a significant limitation in certain applications.
Computational Resources: The paper lacks detailed information on the computational resources required for training and inference, which is crucial for understanding the practical implications of using MDGEN in various research settings.

**Questions:**

How can the model be adapted or improved to reduce its reliance on key frames?

Exploring techniques for unconditional generation or alternative ways to handle the roto-translations without predefined key frames could enhance the model's flexibility and applicability.
What architectural changes or enhancements could improve the model's performance on larger molecular systems such as proteins?

Investigating more scalable architectures or hybrid approaches that combine the current method with other techniques tailored for large systems could address this limitation.
How does the computational cost of training the model compare to traditional MD simulations, and what are the implications for its practical use?

Providing detailed information on computational requirements and potential optimizations could help in assessing the model's feasibility for widespread use.
What alternative tokenization strategies could be explored to extend the model's applicability to a wider range of molecular systems?

Research into tokenization methods that can handle diverse molecular structures and dynamics could broaden the model's utility.
How can additional conditioning types (e.g., textual descriptions, experimental data) be incorporated into the model, and what benefits might they provide?

Experimenting with and integrating various forms of conditioning could enhance the model's ability to generate more accurate and contextually relevant trajectories.
What are the potential impacts of data quality and availability on the model's performance, and how can these challenges be mitigated?

Addressing data-related challenges through techniques like data augmentation, transfer learning, or synthetic data generation could improve the model's robustness and applicability.
Can additional evaluation metrics be developed to provide a more comprehensive assessment of the generated trajectories' quality?

Identifying and implementing new evaluation criteria could offer deeper insights into the strengths and limitations of the model's output.

**Limitations:**

Reliance on Key Frames:

The model relies on key frames for parameterizing the roto-translations, which means it cannot perform unconditional generation or inpainting of residue roto-translations. This dependency might limit its applicability to scenarios where key frames are not easily obtainable or where full trajectories need to be generated from scratch.
Scalability to Larger Systems:

The architecture shows weaker performance on larger systems such as proteins compared to smaller peptides. This suggests that the current model and architecture might not be well-suited for handling the complex motions and larger size of protein structures without further modifications or enhancements.
Computational Resources:

While the paper mentions significant speedups compared to traditional MD simulations, the computational resources required for training the model (e.g., GPU hours) are not explicitly discussed. This information is crucial for understanding the practicality and scalability of the approach.

Generalization to Diverse Systems:

The current tokenization and modeling strategies are tailored to peptides and proteins. For more diverse molecular systems such as organic ligands, materials, or explicit solvent systems, alternative tokenization strategies might be necessary. This limits the immediate applicability of the model to a broader range of molecular simulations.
Limited Exploration of Additional Conditioning:

The paper primarily explores conditioning on initial frames and residue identities. Other types of conditioning, such as textual or experimental descriptors, are not explored but could open up further applications and improve the model's utility.
Data Availability and Quality:

The success of the model heavily depends on the availability of high-quality MD trajectory data. For many complex systems, obtaining such data can be challenging, which could limit the model's applicability and performance.
Evaluation Metrics:

While the paper uses several rigorous metrics for evaluation, the choice of metrics may not fully capture all aspects of the generated trajectories' quality. Additional metrics or more diverse evaluation criteria could provide a more comprehensive assessment.

---

> ### Author Rebuttal · Authors · 2024-08-03
>
> Thank you for the review! To address your questions and concerns:
>
> ---
>
> **Complexity and accessibility**
>
> We have aimed to provide a clear and reproducible method and exposition accessible to the average reader familiar with molecular machine learning. We aimed to make modeling choices that were as simple as possible, i.e.,
> * using a simple stochastic interpolants framework rather than diffusion
> * using a vanilla unmodified transformer from previous work rather than complex frame- or pair-based architectures
> * generating invariant tokens over Euclidean space rather than diffusing over the space of frames and tokens.
>
> To promote reproducibility, we have
> * provided pseudocode for our architecture in Appendix A
> * provided experimental details sufficient for replications and conceptual justifications for design choices and metrics in Appendix B
> * will provide full source code in the revision
>
> While our work bridges molecular dynamics and generative models, it should not require a deep background in either as we have distilled the essential aspects of the relevant literature in Section 2. With that said, if there are any specific aspects of the method that remain unclear despite our best efforts, please let us know.
>
> **Reliance on key frames**
>
> While it is true that the use of key frames impacts the ability to do unconditional generation, we nevertheless opted for this design choice as unconditional generation of trajectories is (to our knowledge) not a problem of scientific interest. On the other hand, the key frames allow the conditional modeling problems to be consideribly simplified and clarified, leading to the strong results shown in our experiments. Hence, the use of key frames could instead be considered a technical insight of our model that enables it to contribute to the development of real-world, impactful scientific problems, at the cost of problems of lesser interest.
>
> **Larger and more diverse systems**
>
> We have focused on tetrapeptides as model systems in this work for two key reasons:
> * We can run simulations for thousands of systems in order to properly test the generalization abilities of our model.
> * We can build Markov State Models for each system, allowing the careful benchmarking for forward simulation and interpolation capabilities.
>
> Both of these aspects are important for the thorough, careful benchmarking of the new capabilities we demonstrate. We opted to prioritize these careful studies to support the core claims of our work, in lieu of expanding its scope to larger and more diverse systems, which we think are best left to future work.
>
> **Conditioning**
>
> We agree that further types of conditioning would be exciting to explore. However, the conditional settings we have already provided represent significant conceptual shifts relative to existing work in learning surrogate models of molecular dynamics. Additional types of conditioning would further expand the technical scope of the work, and the ones we mention (i.e., text conditioning) are highly speculative with large uncertainty with regards to scientific utility, at least at present. With that in mind, it is not clear to us which additional settings are being referenced as weaknesses or limitations. If there are specific areas in mind, please let us know.
>
> **Data availability and quality**
>
> We acknowledge that MD data is required to train the method, and obtaining long trajectories can be time-consuming. However, compared to molecular ML modalities requiring _experimental_ data, such as those that train on crystal structures, it is much easier to obtain high-quality data by running simulations than by running wet-lab experiments. Hence, works like ours that train on _simulated_ data actually help  mitigate the challenges of data availability and quality in molecular machine learning.
>
> **Evaluation metrics**
>
> We have designed and implemented thorough and principled metrics that assess the ability of our model to produce trajectories:
> * With good distributional properties over structures, assessed by the Jensen-Shannon divergences and Markov state occupancies
> * With good dynamical properties over motions, assessed by the various transition path metrics, the decorrelation times, flux matrix correlations, autocorrelation functions, and dynamical content.
>
> While other metrics are certainly possible, it is not clear to us which areas are concretely being referenced as weaknesses or limitations of the current evaluations. If there are specific areas in mind, please let us know.
>
> **Computational resources**
>
> We report our training time for each model as follows, measured on A6000 GPUs:
>
> * Forward simulation: 412 GPU-hrs
> * Interpolation: 272 GPU-hrs
> * Upsampling: 292 GPU-hrs
>
> We have reported inference time comparisons with MD simulations in Table 2 on the forward simulation tasks, all on one A6000 GPU. Our method is much faster than the baseline simulations, and much more accurate than an abridged simulation of the same wall-clock time.
>
> Emphatically, the training time is amortized across all systems since our method is _transferable_ by design. That is, for the fixed training cost we obtain a set of models that can be applied, with the reported accuracy, to _any_ tetrapeptide system at test time. Thus, our model training offers a clear and overwhelming advantage over running long MD simulations for individual tetrapeptide systems.
>
> ---
>
> We hope the new discussion and results address your concerns! Please let us know if there are further opportunities to improve the score.

---

> > ### Comment · Reviewer_TTLp · 2024-08-12
> > **Thanks for the clarifications**
> >
> > Some of my concerns have been addressed. I raised my score.

---

### Official Review · Reviewer_QAiG · 2024-07-24

**Soundness:** 4
**Presentation:** 3
**Contribution:** 4
**Rating:** 7
**Confidence:** 4

**Summary:**

The authors introduce MDGen as a novel approach for modeling MD trajectories. They demonstrate the capabilities of this method in tasks such as interpolation, upsampling, and inpainting of small peptides. The accuracy as well as speed of the new approach compared to the ground truth baseline is quantitatively evaluated. Initial experiments toward upscaling to small proteins are shown.

**Strengths:**

The idea of MDGen is novel and very well presented in this manuscript. The results are convincing and interesting.

**Weaknesses:**

1. Parts of Sections 3.1 and 3.2 are very condensed and hard to follow. A more detailed description in the SI would be helpful, where the most important aspects of the cited work is also repeated.
2. The suitability of the chosen representation for longer amino acid chains is questionable. This is also mentioned in the manuscript, but nonetheless, proteins are mentioned many times (more than 30) in the manuscript, while almost all experiments are actually performed on very small peptides. It should be stated in a more prominent place that upscaling to proteins is not trivial.
3. The representation limits the model to learn MD trajectories of natural amino acids, as no all-atom representation is used directly. This should be made clearer in the manuscript.

Minor points: A lot of figures have no proper axis labels (e.g. Fig 3, 4, 5, 6). This should be fixed. The best models in Table 4 should be indicated in bold.

**Questions:**

1. How often do clashes and other high-energy structures occur in the generated trajectories?
2. When comparing to other methods and approaches in the experimental section - do all of them use a similar reduced representation or do the other methods generate all-atom representations?

**Limitations:**

The approach is limited to peptides. Transfer to any other molecules is questionable due to a lack of suitable representation/tokenization.

---

> ### Author Rebuttal · Authors · 2024-08-03
>
> Thank you for the review! To address your questions and concerns:
>
> ---
>
> **Parts of Section 3 hard to follow**
>
> Due to space limitations, the description of our method in Section 3 was indeed a bit condensed. We will expand the exposition with the extra page allotted in the revision.
>
> **Suitability for proteins and non-AA molecules**
>
> We sought to qualify our results on protein simulations and apologize if a different impression was conveyed. In the revision we will increase emphasis that the focus of the paper is on peptides and remove prominent mention of proteins from the abstract and introduction, as suggested.
>
> With that said, we have focused on tetrapeptides as model systems in this work for two key reasons:
> * We can run simulations for thousands of systems in order to properly test the generalization abilities of our model.
> * We can build Markov State Models for each system, allowing the careful benchmarking for forward simulation and interpolation capabilities.
>
> Both of these aspects are important for the thorough, careful benchmarking of the new capabilities we demonstrate. We opted to prioritize these careful studies to support the core claims of our work, in lieu of expanding its scope to larger and more diverse systems, which we think are best left to future work.
>
> **Clashes in the generated structures**
>
> To assess the frequency of clashes or high-energy structures in MDGen forward simulation rollouts, we compute the distributions of
> * The closest distance between nonbonded atoms
> * Nonbonded energy (Coulomb + Lennard-Jones)
> * Torsional energy
> * Heavy atom bond lengths
> * Radius of gyration
>
> These distributions are shown and compared to the ground truth in **Figure 1** in the PDF attached to the global response. We find that the vast majority of MDGen structures are of high quality (i.e., clashes are rare) and adhere closely to the ground truth distributions.
>
> **Representations of other methods**
>
> The peptide representations of the other methods are summarized as follows:
>
> * For most evaluations we compare to Markov state models, which emit discretized representations of the trajectory. These are much more coarse-grained than our reduced representation.
> * AlphaFlow emits residue frames and torsion angles, an equivalent representation to ours.
> * The inpainting baselines are inverse folding models and do not emit trajectories of any kind.
> * Ground truth molecular dynamics provides all-atom trajectories.
>
> **Minor points**
>
> Thank you for noting these; we will incorporate them in the revision.
>
> ---
>
> We hope the new discussion and results address your concerns! Please let us know if there are further opportunities to improve the score.

---

> > ### Comment · Reviewer_QAiG · 2024-08-08
> >
> > Thank you for the clarifications. This answers all my questions and I encourage that all updates announced here will be integrated in the final version. As the changes do not change the main message and claims, I will keep my score as it is, so I continue to support acceptance of the paper.

---

### Author Rebuttal · Authors · 2024-08-03

# Overall Response

We thank all reviewers for their time taken in providing constructive feedback!

In addition to the individual responses, we also provide new **figures and visualizations in the PDF file** attached to this global response.

* In Figure 1, we compare the distributions of additional **structural and energy metrics** (e.g., clashes) from MDGen rollouts and from reference MD simulations, as suggested by **Reviewers QAiG** and **hVZ7**. We find that the vast majority of MDGen structures are of high quality and closely resemble the ground truth distribution on these metrics.
* In Figures 2 and 3, we provide **comparisons with Timewarp and ITO** as requested by **Reviewers zaG3** and **b8Jp**. In Figure 2, we show that Timewarp and ITO suffer from unstable inference rollouts out-of-the-box on generic tetrapeptides. However, we correct for these instabilities ourselves to obtain a meaningful comparison. In Figure 3, we visualize free energy surfaces and torsion angle distributions obtained from Timewarp and ITO compared with MDGen. We find that MDGen obtains better results across the board.
* In Figure 4, we visualize an **interpolation path between apo / holo states** of adenylate kinase (1AKE / 4AKE) as suggested by **Reviewer hVZ7**, obtained from a newly trained ATLAS interpolation model.

---

### Decision · Program_Chairs · 2024-09-25

**Decision:**

Accept (poster)

**Comment:**

The paper presents a new approach to a timely topic -- modeling molecular dynamics -- with two accepted papers in NeurIPS last year. There are several contributions, lending ideas from CV and NLP (such as masking and impainting) to model a joint distribution of the simulation trajectory. As such, these new training strategies allow for the use of more data than previous approaches. The tasks the authors explore are in line with standard benchmarks. New prediction tasks are introduced, mostly of limited interest and practical use (e.g. trajectory upsampling). The major limitation and strength of the approach are its main innovations: 1) the molecular representation (tokenization), which locks in the models to be proteins or peptides, with no clear strategy to generalize to molecules more broadly, and 2) the trajectory representation, requiring end users to make hard decisions at training time, which may limit inference time usefulness. The underlying learning objective is not well justified from a physical perspective: e.g. a trajectory is learned as a joint distribution suggesting a time-dependence of the generator. There are settings where this is the case, however, in all the systems explored the paper the generating process is homogeneous and Markovian, and therefore at odds with the model and training setup. Nevertheless, the performance appears good in certain cases (small peptides), but poor on larger scale application, e.g. ATLAS protein dataset.